# Energetic and Protective Coating via Chemical and Physical Synergism for High Water-Reactive Aluminum Powder

**DOI:** 10.3390/ma15238554

**Published:** 2022-12-01

**Authors:** Lichen Zhang, Shuo Wang, Lixiang Zhu, Xiaodong Li, Xing Su, Meishuai Zou

**Affiliations:** School of Materials Science and Engineering, Beijing Institute of Technology, Beijing 100081, China

**Keywords:** aluminum, energetic, intact coating, hydrophobicity

## Abstract

Aluminum powder plays important role in the field of energetic materials. However, it is often vulnerable to oxygen and water due to the high reactivity of aluminum, and it is challenging to build up uniform and passivated coating via existing means. In this work, (Heptadecafluoro-1,1,2,2-tetradecyl) trimethoxysilane (FAS-17) and glycidyl azide polymer (GAP) were used to coat the surface of high water-reactive aluminum powder (w-Al) to form inactivated w-Al@FAS-17@GAP energetic materials, via the synergy of chemical bonding and physical attraction. Thermal reaction tests showed that the exothermic enthalpy of w-Al@FAS-17@GAP was 5.26 times that of w-Al. Ignition tests showed that w-Al@FAS-17@GAP burnt violently at 760 °C, while w-Al could not be ignited even at 950 °C. In addition, the combined coating of FAS-17 and GAP could effectively improve the hydrophobicity and long-term stability of w-Al, which helped to overcome the poor compatibility of w-Al with explosive components. Our work not only displayed an effective routine to synthesize O_2_/H_2_O proof Al energetic materials, but also pointed out a synergistically chemical and physical strategy for constructing intact high-performance surfaces.

## 1. Introduction

Aluminum powder plays an important role in energetic materials, due to its high combustion calorific value, low price, and high energy density [1,2,3,4]. At the same time, Aluminum is the most abundant crustal metal on Earth and can be fully recycled, which gives it good potential for hydrogen production [5,6,7]. In order to achieve efficient and fast hydrogen production from aluminum at room temperature or low temperature, researchers prepared aluminum-based low-melting point metal alloys [8,9,10]. This kind of high water-reactive Al (w-Al) provide the possibility for the multistage energy release of underwater explosives. However, it is highly reactive to water, susceptible to oxidation, and poorly compatible with explosive components. Therefore, its adaptability in the environment during storage is poor, which brings great challenge to the manufacture of explosive column. It could not be directly added to explosives as metal fuels.

In order to increase the compatibility and environmental adaptability between high water-reactive Al and explosive components, it needs to be coated and passivated. To improve stability against oxidation of aluminum powder in storage, hydroxyl terminated polybutadiene–toluene diisocyanate (HTPB-TDI) was used to coat aluminum powder [11]. Cao, et al. [12] coated an aluminum–lithium alloy by in situ polymerization of styrene to improve stability. Tran, et al. [13] coated (3-aminopropyl) triethoxysilane on the surface of aluminum powder to improve the oxidation resistance and stability in water. However, these protective coatings had the potential of reducing the combustion properties of aluminum powder. Fluorides were mostly studied to coat on the surface of Al, such as poly(vinylidene fluoride) (PVDF), pentadecafluorooctanoic acid (PFOA), ammonium perfluorooctanoate (APFO), fluorographene (FG), and polydopamine fluoride (PF) [14,15,16,17,18,19]. The decomposition products of fluoride undergo a pre-ignition reaction with aluminum, which promotes the combustion of aluminum powder. Therefore, the combustion efficiency and energy release efficiency can be improved [14,15,20]. Due to the water reactivity of w-Al, the pretreatment of weak acid or base aqueous solution used for coating PVDF is not suitable, and fluorides such as perfluoric acid cannot be considered, as it can react with w-Al. By comparison, fluorine-containing silanes could be good choices for passivation coatings. Jiang, et al. [21] applied 1H, 1H, 2H, 2H-perfluorooctyltriethoxysilane on the surface of aluminum powder to improve the ignition and combustion performance of aluminum powder. However, 1H, 1H, 2H, 2H-perfluorooctyltriethoxysilane has low fluorine content, low boiling point, and the preparation process is volatile. (Heptadecafluoro-1,1,2,2-tetradecyl) trimethoxysilane (FAS-17) was used in the construction of hydrophobic layers [22,23,24]. It was not volatile, with the boiling point as high as 203 °C, and it can be coated on the surface of w-Al to form a passivation protection layer. However, FAS-17 is not an energetic material; the FAS-17 coating will significantly reduce the energy level of the aluminum powder system. Glycidyl azide polymer (GAP) is an energetic binder commonly used in explosives and propellants [25,26]. Another layer of GAP coating can not only increase the energy of the aluminum powder system, but also improve the compatibility with the explosive components [2]. However, GAP is highly polar, and incompatible with conventional fluorides [2]. Unlike the fluorides containing polar functional groups including -OH, -COOH, -NH_2_, and etc., highly hydrophobic siloxane groups further contribute to the non-polarity of fluorine-containing silanes [27]. This is challenging for the combined use of fluorine-containing silanes and GAP, which is seldom reported.

In this paper, we proposed an energetic and protective coating for high water-reactive aluminum powder. FAS-17 was coated on w-Al to obtain w-Al@FAS-17 mainly through the covalent bonds formed between Si-O groups in FAS-17 and Al-OH groups in w-Al, as well as the high electron affinity between Al and F [2,28]. GAP was then covalently linked to Si-O by C-OH, forming w-Al@FAS-17@GAP. The thermal reaction properties, compatibility, and environmental adaptability were tested by thermogravimetric analysis (TGA) and differential scanning calorimetry (DSC). The hydrophobicity and hydrogen production tests showed that the w-Al had been highly passivated. Oxygen bomb calorimeter and ignition tests proved that w-Al@FAS-17@GAP could be used as energetic metal fuel in explosives. The hydrophobicity and stability in storage of w-Al was significantly enhanced, and the corresponding w-Al@FAS-17@GAP was not reactive with water. The uniformly compounded fluoride/GAP composite coating displayed a synergistic effect on improving the ignition and combustion properties of w-Al. The exothermic enthalpy of w-Al@FAS-17@GAP could be enhanced by over four times, which was remarkable among the existing energetic Al materials. Our work not only proposes an effective routine for a high-performance protective and energetic coating for Al, but also points out an inspiring methodology for constructing uniform and stable functional coatings. This strategy may have potential applications in military uses, aerospace, heterogeneous membrane, ultrathin surface modification, metal protection, intelligent composite coating, etc.

## 2. Experimental

### 2.1. Materials

High water-reactive aluminum powder (w-Al, D50 = 50 μm), Al, Ga, In, and Sn metals were mixed in a 94:3.8:1.5:0.7 mass ratio and melted together in a resistance furnace at 900 °C for 1 h under Ar. The resulting casts were mechanically crushed and sieved to obtain w-Al following the method in our previous work [10]. (Heptadecafluoro-1,1,2,2-tetradecyl) trimethoxysilane (FAS-17, purity: 98.0%), tetrahydrofuran (THF, AR) and deionized water were purchased from Beijing Innochem Science & Technology co. (Beijing, China), LTD. Glycidyl azide polymer (GAP, n ≈ 20, AR) was acquired from Liming Chemical Engineering Institute (Luoyang, China). All chemicals were used as received.

### 2.2. Samples Preparation

A mass of 2 g of w-Al was poured into 50 mL tetrahydrofuran to form an aluminum suspension. After stirring for 10 min, 0.1 mL deionized water was added dropwise to generate abundant hydroxyl groups on the surface of w-Al. Then 0.5 g FAS-17 was added after stirring for 20 min. Stirring continued for another 3 h. FAS-17 was coated on the surface of w-Al mainly through covalent bonding of siloxane and hydroxyl groups, and the physical electron affinity of Al-F played an auxiliary role in the coating effect. The aluminum powder was filtered, washed, and dried in an 80 °C vacuum oven (Lab companion, Dongguan, China) for 6 h to obtain w-Al@FAS-17. The preparation process of w-Al@FAS-17@GAP was based on w-Al@FAS-17 by adding 0.1 g GAP. GAP was covalently linked to FAS-17 via covalent bonds between excess siloxane in FAS-17 and hydroxyl groups in GAP. w-Al@FAS-17@GAP was also obtained by filtering, washing, and drying. The schematic diagram of the preparation process of w-Al@FAS-17@GAP was shown in Figure 1.

### 2.3. Methodology

The morphology and elemental distribution of w-Al, w-Al@FAS-17 and w-Al@FAS-17@GAP were detected by scanning electron microscopy (SEM) and energy dispersive X-ray spectroscopy (EDS, Hitachi S-4800, Tokyo, Japan). All samples were sputtered on the surface with Au. Fourier transform infrared (FTIR) spectra of FAS-17, GAP, w-Al, w-Al@FAS-17 and w-Al@FAS-17@GAP were conducted with a Nicolet 6700 IR spectrometer (Waltham, MA, USA).

Thermogravimetric analysis (TGA) and differential scanning calorimetry (DSC) of w-Al, w-Al@FAS-17 and w-Al@FAS-17@GAP were conducted on a synchronous thermal analyzer (STA, NETZSCH STA 449 F3, Bavaria, Germany) at a heating rate of 20 °C/min. The measuring range was from 40 °C to 1100 °C under air atmosphere. GAP was mixed with w-Al and w-Al@FAS-17@GAP in a mass ratio of 1:1, and the compatibility between aluminum powder samples and GAP was measured by DSC (heating rate 10 °C/min, from 40 °C to 600 °C under nitrogen atmosphere). The evaluation standard referred to the compatibility standard proposed by Honeywell Company [29,30]. w-Al and w-Al@FAS-17@GAP were placed at standard atmosphere (23 °C, relative humidity: 50%) for one month to evaluate the environmental adaptability of aluminum powder by testing its thermal oxidation properties at air atmosphere. The heating rate was set to be 20 °C/min and the temperature range was set from 40 °C to 1100 °C.

The water reactivity of w-Al, w-Al@FAS-17 and w-Al@FAS-17@GAP was evaluated by measuring the volume of hydrogen produced by the reaction with H_2_O. Samples with a mass of 0.5 g were added in a three-neck flask, and 10 mL water (20 °C) was injected with a syringe, keeping the water/fuel mass ratio at 20:1. No mixing or heating was performed during the reaction process. Hydrogen generation rates were then measured using a mass flow meter (FS4008, MEMS Technologies, Beijing, China) based on our previous work [31]. An SL 200B control system was applied for the static water drop contact angle measurement to explore facial hydrophobicity of w-Al, w-Al@FAS-17 and w-Al@FAS-17@GAP.

The combustion heats of w-Al, w-Al@FAS-17 and w-Al@FAS-17@GAP were obtained with an oxygen bomb calorimeter (BCA 500, IDEA, San Diego, CA, USA). Each sample was tested three times to obtain an average value. The combustion processes of w-Al, w-Al@FAS-17 and w-Al@FAS-17@GAP were tested by a high temperature melting furnace equipped with high-speed photography with a sampling rate of 960 frames/s. Each sample (~0.2 g) was weighed and tested three times to ensure reliability. The products of the ignition tests were analyzed by XRD.

## 3. Results and Discussion

### 3.1. Morphologies and Composition

The morphologies and elemental composition of w-Al, w-Al@FAS-17 and w-Al@FAS-17@GAP revealed by SEM-EDS images were shown in Figure 2. From Figure 2a, the morphology of w-Al was irregular and there were many cracks on the surface, which was a typical sign of its high reactivity with water [10]. When a sufficient amount of FAS-17 was coated, the surface of w-Al@FAS-17 no longer had cracks and was slightly smooth, forming irregular coating layers. When GAP was further coated on w-Al@FAS-17, the surface smoothness of w-Al@FAS-17@GAP was significantly improved. Through EDS mapping images and point scanning images in Figure 2d,e, it was clearly seen that C, O, N, F and Si elements were uniformly distributed on the surface of w-Al@FAS-17@GAP. From Figure 2d–g, the introduction of GAP did not interrupt the original fluoride layer, and the only change was the additional N element. It was preliminarily expected that the Al-FAS and FAS-GAP chemical bonds would facilitate the formation of the organic coating layer with efficiency and uniformity. During coating (Figure 1), FAS-17 was coated on Al, which was dominated by a chemical reaction between the siloxane and hydroxyl groups. At the same time, some “free” FAS-17 molecules were loaded via physical electron affinity among Al-F groups [28], leaving abundant unreacted siloxane functional groups. Afterward, GAP molecules were chemically grafted, forming energetic segments.

FTIR spectra of FAS-17, GAP, w-Al, w-Al@FAS-17 and w-Al@FAS-17@GAP are shown in Figure 3 to further demonstrated that FAS-17 and GAP were coated on w-Al. As shown in Figure 3a, FAS-17 and GAP had several notable peaks in their FTIR. Namely, the details were 2952 cm^−1^ (CH_3_), 1197 cm^−1^ (C-F), 1145 cm^−1^ (Si-O), 3407 cm^−1^ (OH^−^), 2927 cm^−1^ (CH_2_), 2875 cm^−1^ (CH), 2094 cm^−1^ (-N_3_), 1274 cm^−1^ (C-N), and 1074 cm^−1^ (C-O) [2,32]. After coating with FAS-17, w-Al@FAS-17 showed three notable bands at 2971 cm^−1^, 1198 cm^−1^, and 1145 cm^−1^, which were assigned to CH_3_, C-F and Si-O. FAS-17 was coated on the surface of w-Al by eliminating the methyl groups and forming Si-O bonds to connect with the hydroxyl group [21]. The methyl groups of w-Al@FAS-17 still existed, indicating that FAS-17 was excessive. This could provide abundant reactive sites for the subsequent GAP coating. As for w-Al@FAS-17@GAP, the characteristic peak of CH_3_ disappeared, indicating that all “free” FAS-17 were chemically grafted via Si-O bonds. A strong characteristic peak (2095 cm^−1^) of azide appeared, which also proved the successful coating of GAP. Combining SEM-EDS and FTIR spectra, we found that FAS-17 was densely coated on the surface of aluminum powder through Al-O-Si (chemical bonding) and electron affinity among Al-F groups (physical bonding), and GAP was further coated by combining with FAS-17 through Si-O bonding, forming a dual uniform coating structure.

### 3.2. Thermal Reaction Properties

Simultaneous thermal analysis was used to detect the thermal reaction performance of aluminum powder with air [28]. From Figure 4a, the mass of w-Al increased slowly with enhancing temperature in the air atmosphere, with an ultimate increase ratio of 5.18% at 1100 °C. Meanwhile, w-Al@FAS-17 and w-Al@FAS-17@GAP went through three stages. In the first stage, mass loss occurred, which was caused by the thermal decomposition of FAS-17 and GAP at 100–500 °C. Next, the mass of w-Al@FAS-17 and w-Al@FAS-17@GAP stayed relatively stable at 500–700 °C. Finally, obvious weight gain was achieved due to the oxidation of aluminum powder at 700–1100 °C. The weight gains of w-Al@FAS-17 and w-Al@FAS-17@GAP were 6.44% and 8.03%, respectively. Compared with w-Al, the increase in weight gain indicated that the thermal oxidation efficiency of w-Al@FAS-17@GAP was improved. The wide exothermic peak in the DSC curve of w-Al@FAS-17 was caused by the decomposition of FAS-17. The two exothermic peaks in the DSC curve of w-Al@FAS-17@GAP were the thermal decomposition of FAS-17 and GAP, respectively. The temperatures corresponding to the endothermic peaks of w-Al, w-Al@FAS-17 and w-Al@FAS-17@GAP were similar. They were around 652 °C, which were the melting peaks of aluminum. At 700–1100 °C, the area surrounded by the DSC curve and the baseline was the exothermic enthalpy of aluminum powder oxidation. The exothermic enthalpy of w-Al was only 231.9 J/g, while the exothermic enthalpy of w-Al@FAS-17@GAP was as high as 1219.3 J/g, which was 4.26 times higher than that of w-Al. This realized the high energy release of w-Al@FAS-17@GAP.

### 3.3. Compatibility and Environmental Adaptability

Good compatibility with explosive components is an important prerequisite for the application of aluminum powder in explosives [33,34]. Therefore, it is necessary to evaluate the compatibility of our proposed aluminum powder system with explosive components. We used DSC to test the difference in decomposition temperature of GAP, GAP/w-Al and GAP/w-Al@FAS-17@GAP to judge the compatibility. From Figure 5, the temperature of the exothermic peaks (T_p_) of GAP, GAP/w-Al and GAP/w-Al@FAS-17@GAP were 252.5 °C, 245.8 °C and 251.5 °C, respectively. ΔT_p_ is the T_p_ of GAP minus T_p_ of mixed aluminum powder. For GAP/w-Al, ΔT_p_ was 6.7 °C, indicating that GAP and w-Al were sensitized and poor compatibility was addressed. The ΔT_p_ of GAP/w-Al@FAS-17@GAP was 1.0 °C, indicating that GAP and w-Al@FAS-17@GAP had good compatibility. Therefore, it was realized that the compatibility of w-Al could be improved by a coating of FAS-17 and GAP.

In addition, environmental adaptability is also one of the important issues for the storage of aluminum powder. It could be seen from Figure 6 that the thermogravimetric curve of the contrast sample, abbreviated as w-Al (E), was obviously different from that of w-Al. When the temperature was raised from 40 °C to 660 °C for w-Al (E), the corresponding mass loss rate was 5.15%. The corresponding weight gain was only 1.74% at 700–1100 °C, indicating that the content of active Al was reduced due to oxidation and hydration. Such strong hygroscopicity and poor environmental adaptability of w-Al makes it challenging for storage. Distinctively, the TGA curve of w-Al@FAS-17@GAP (E) was similar to that of w-Al@FAS-17@GAP, with a similar weight gain of 8.10% and 8.03%, respectively, at 700–1100 °C. This showed that the environmental adaptability of w-Al was significantly improved after coating with chemically bonded FAS-17 and GAP.

### 3.4. Water Reactivity

The water reactivity of w-Al was crucial in practice. The water reactivity of aluminum powders before and after coating was evaluated from two aspects: hydrogen production and hydrophobicity. Figure 7a showed the hydrogen generation of equal weight (1 g) w-Al, w-Al@FAS-17, and w-Al@FAS-17@GAP in H_2_O at 20 °C. The hydrogen production rate of w-Al was fairly high in the early stage. Afterward, the reaction with water was slowed down after 200 s, and then basically stopped at 540 s. The average hydrogen generation rate of w-Al was 1.7 mL/(g·s). Whereas, w-Al@FAS-17 and w-Al@FAS-17@GAP did not react with water and no hydrogen was produced, indicating excellent stability in water. The powders of w-Al, w-Al@FAS-17, and w-Al@FAS-17@GAP were pressed into pieces for water static contact angle tests. w-Al reacted with water violently once in contact, resulting in a large number of bubbles. Therefore, the water contact angle of w-Al could not be measured, as shown in Figure 7b. Distinguishingly, the static water contact angle of w-Al@FAS-17 and w-Al@FAS-17@GAP were measured to be 124.5° and 129.5°, respectively. After coating with FAS-17, w-Al turned from high water-reactive to hydrophobic. The contact angle was further increased by 5° after coating, even with highly polar GAP. This indicated that with the FAS-17 layer in advance, the subsequent addition of GAP induced the phase separation in the coating. Despite the chemical bonds between FAS-17 and GAP, their intrinsically physical incompatibility caused the FAS-17 phase to form more compact microstructure. Such a self-hydrophobization effect further improved the hydrophobicity of the coating for w-Al@FAS-17@GAP [35]. Furthermore, we provided a simple set of comparative videos to demonstrate the chemical inertness of w-Al@FAS-17@GAP to water. Appendix A illustrated that w-Al reacted rapidly with water and produced a large number of bubbles. In contrast, w-Al@FAS-17@GAP powder quickly spread on water surface and did not react with the water at room temperature, as shown in Appendix A. In conjunction with the environmental adaptability and water reaction performance of the aluminum powder samples, it was found that the w-Al covered by FAS-17 and GAP had good stability in both air and water, avoiding oxidation and moisture absorption.

### 3.5. The Ignition and Combustion Performance

The combustion behaviors of w-Al, w-Al@FAS-17 and w-Al@FAS-17@GAP in O_2_ atmosphere (3 atm) were measured by oxygen bomb calorimeter. In order to simulate the instantaneous high-temperature reaction of w-Al, w-Al@FAS-17 and w-Al@FAS-17@GAP, 0.2 g samples were instantly poured into the high temperature melting furnace at different temperatures. The lowest ignition temperature and burning duration were observed. As shown in Table 1, the experimental combustion heat values of w-Al, w-Al@FAS-17 and w-Al@FAS-17@GAP were 26.9 kJ/g, 24.8 kJ/g and 26.1 kJ/g, respectively. Due to the effective coating of non-energetic FAS-17, the combustion heat of w-Al@FAS-17 was inevitably decreased by 7.8%, compared with w-Al. After coating with energetic GAP, the combustion heat of w-Al@FAS-17@GAP was increased by 5.2% compared with w-Al@FAS-17, showing enhanced energy level. The combustion heat is an important indicator for aluminum-based fuels. We compared the combustion heat values in other literatures in Figure 8 and found that the combustion heat value of w-Al@FAS-17@GAP was at the medium level of published literatures, and could meet the energy requirements of aluminum-based fuels. More importantly, considering that others seldom used w-Al because of its high instability, our work offered the possibility of utilizing w-Al in the application of energetic Al fuels. Figure 9 contains snapshots of the ignition tests for w-Al, w-Al@FAS-17 and w-Al@FAS-17@GAP. As shown in Figure 9a, w-Al was melted at 950 °C for 3.3 s, but failed to ignite even after 15 s. w-Al@FAS-17 could be ignited at 790 °C after a delay time of 1.7 s, and continued to burn for 2.3 s. This was because the thermal decomposition products of FAS-17 reacted with aluminum or aluminum oxide, leading to a typical pre-ignition phenomenon [14]. As for w-Al@FAS-17@GAP, the ignition delay time was further shortened to 0.3 s, and there were two ignition stages at 760 °C. Distinct from w-Al@FAS-17, the first-stage ignition phenomenon was due to the thermal decomposition of GAP at a relatively low temperature, which promoted the combustion of aluminum powder. The flame at 1.8 s was due to the promoting effect of FAS-17, and the total burning duration of w-Al@FAS-17@GAP was 3.0 s. It can be seen from Figure 9d that after ignition test of w-Al, only a small amount of α-Al_2_O_3_ and γ-Al_2_O_3_ was detected due to its unsuccessful combustion [36]. After combustion, not only was some α-Al_2_O_3_ generated in w-Al@FAS-17, but also two new peaks of AlF_3_ appeared. This was due to the pre-ignition effect of FAS-17. As for w-Al@FAS-17@GAP, more peaks of Al_2_O_3_ and AlF_3_ appeared, and the combustion reaction of w-Al@FAS-17@GAP was greatly promoted. Clearly, w-Al@FAS-17@GAP had a great improvement over w-Al in terms of ignition and combustion properties.

**Figure 8 materials-15-08554-f008:**
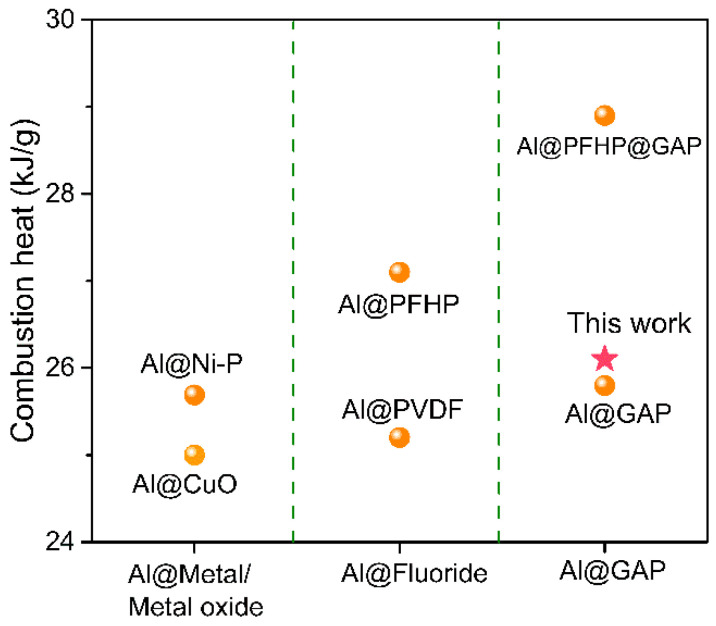
The comparison of combustion heat of this work with other literatures (Al@Ni-P [37], Al@Cuo [38], Al@PFHP [39], Al@PVDF [38], Al@PFHP@GAP [2], Al@GAP [40].

## 4. Conclusions

In this study, FAS-17 and GAP were successfully coated on the surface of w-Al by multiple covalent bonding to prepare energetic and environmentally durable aluminum powder (w-Al@FAS-17@GAP). Compared with w-Al, the reactivity of w-Al@FAS-17@GAP via thermal oxidation was improved, and the exothermic enthalpy was over four times higher than that of w-Al. In addition, the dual coating of FAS-17 and GAP improved the compatibility between w-Al and the effective components in explosives, enhanced the stability upon long-term storage, and encouraged hydrophobicity with great inertness to water. As for ignition and combustion behaviors, greatly promoted ignition and combustion processes were achieved for w-Al@FAS-17@GAP, while w-Al could not even be ignited at 950 °C. Therefore, this study provides a new method for the preparation of high-energy protective coatings on w-Al surfaces with synergistic Si-O bond (chemical interaction) and strong electron affinity (physical interaction), which holds a lot of promise in highly stable and energetic w-Al fuels, explosives, and propellants. In addition, our work may be instructive for other future metallic surface engineering, such as heterogeneous self-assembly, superhydrophobic coating, ultrathin facial patterning, etc.

## Figures and Tables

**Figure 1 materials-15-08554-f001:**
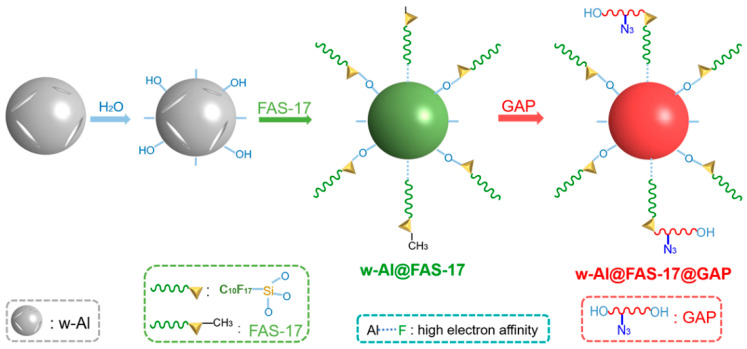
The schematic diagram of the preparation process of w-Al@FAS-17 and w-Al@FAS-17@GAP.

**Figure 2 materials-15-08554-f002:**
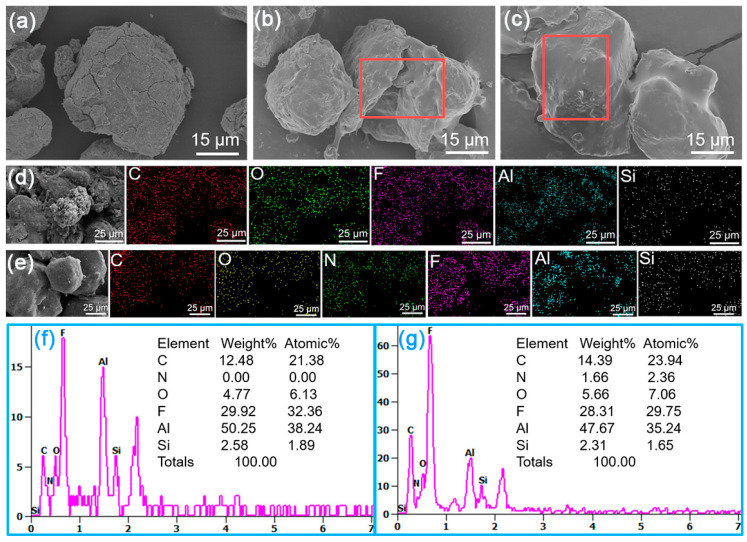
The SEM images of w-Al (**a**), w-Al@FAS-17 (**b**) and w-Al@FAS-17@GAP (**c**); EDS mapping images of w-Al@FAS-17 (**d**) and w-Al@FAS-17@GAP (**e**); EDS point scanning images of w-Al@FAS-17 (**f**) and w-Al@FAS-17@GAP (**g**).

**Figure 3 materials-15-08554-f003:**
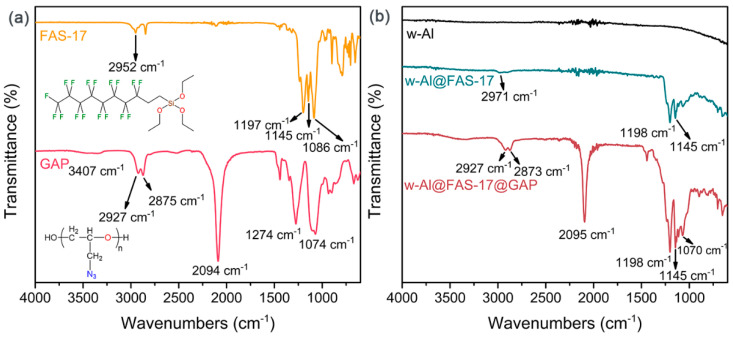
FTIR spectra of FAS-17, GAP (**a**) and w-Al, w-Al@FAS-17, w-Al@FAS-17@GAP (**b**). (2971–2952 cm^−1^, CH_3_; 1197 cm^−1^, C-F; 1145 cm^−1^, Si-O; 1086–1070 cm^−1^, C-O; 3407 cm^−1^, OH^−^; 2927 cm^−1^, CH_2_; 2875 cm^−1^, CH; 2094 cm^−1^, -N_3_; 1274 cm^−1^, C-N).

**Figure 4 materials-15-08554-f004:**
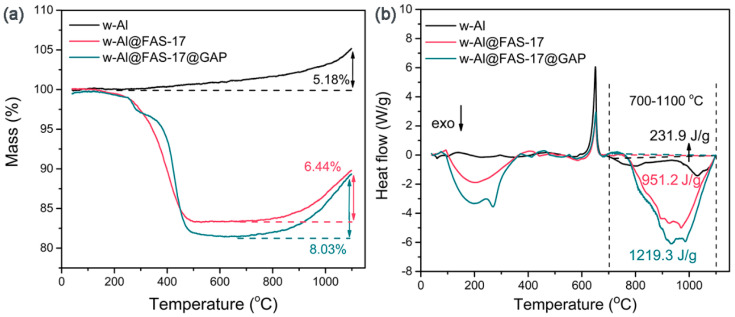
TGA curves with comparison of weight gains (**a**), DSC curves with exothermic enthalpy energy at 700–1100 °C (**b**) of w-Al, w-Al@FAS-17 and w-Al@FAS-17@GAP.

**Figure 5 materials-15-08554-f005:**
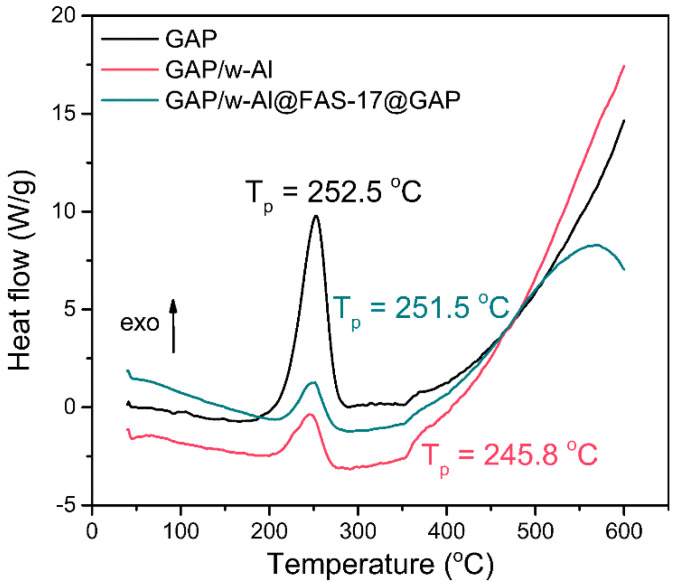
DSC curves of GAP, GAP/w-Al and GAP/w-Al@FAS-17@GAP.

**Figure 6 materials-15-08554-f006:**
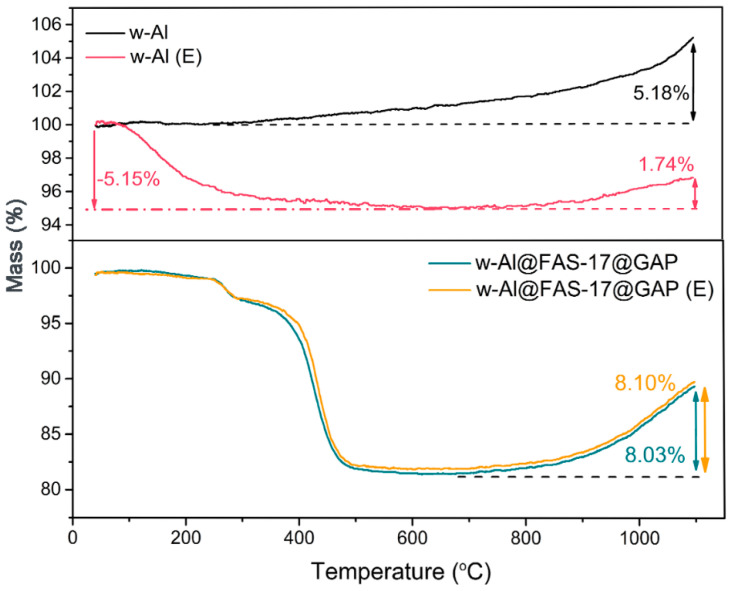
Comparison of TGA curves for environmental adaptability of w-Al and w-Al@FAS-17@GAP.

**Figure 7 materials-15-08554-f007:**
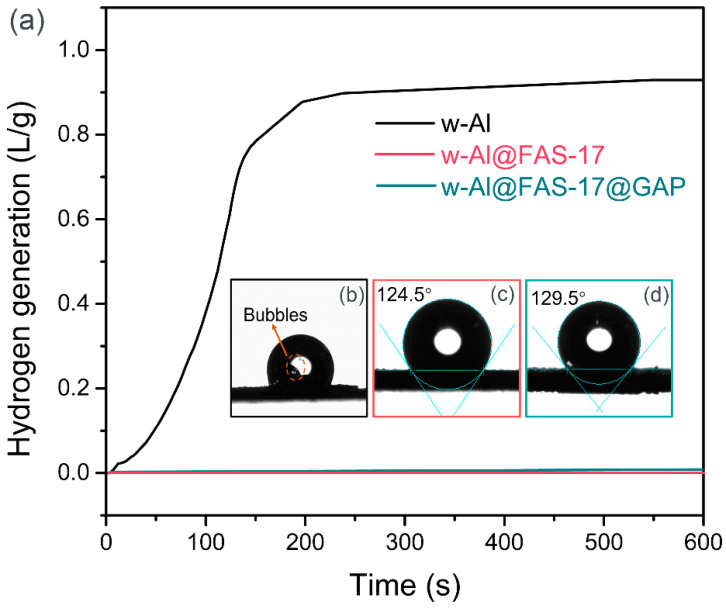
Hydrogen generation curves of w-Al, w-Al@FAS-17 and w-Al@FAS-17@GAP in H_2_O (**a**), photographs of static water contact angle of w-Al (**b**), w-Al@FAS-17 (**c**) and w-Al@FAS-17@GAP (**d**).

**Figure 9 materials-15-08554-f009:**
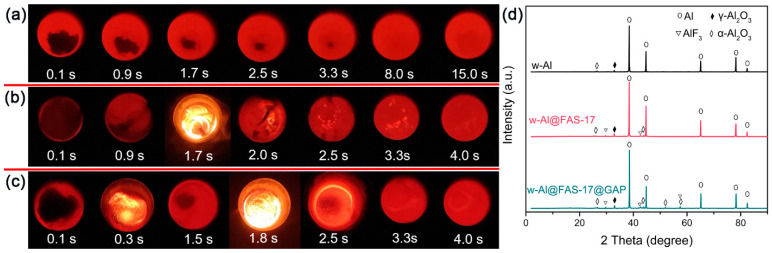
Snapshots of ignition test of w-Al (**a**), w-Al@FAS-17 (**b**) and w-Al@FAS-17@GAP (**c**); XRD patterns (**d**) of the ignition products of w-Al, w-Al@FAS-17 and w-Al@FAS-17@GAP.

**Table 1 materials-15-08554-t001:** The data of combustion heat and ignition test.

Sample	Oxygen Bomb Calorimeter Test	Ignition Test		
	Heat Release (kJ/g)	Minimum Ignition Temperature (°C)	t_I_^*^ (s)	t_B_^#^ (s)
w-Al	26.9	--	--	--
w-Al@FAS-17	24.8	790	1.7	2.3
w-Al@FAS-17@GAP	26.1	760	0.3	3.0

t_I_^*^ ignition delay time at the lowest ignition temperature. t_B_^#^ burning duration at the lowest ignition temperature. -- this data was unavailable as w-Al cannot be ignited at 950 °C.

## Data Availability

Not applicable.

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
