# Peer review of "Energetic and Protective Coating via Chemical and Physical Synergism for High Water-Reactive Aluminum Powder"

_materials, 2022, doi:10.3390/ma15238554_

Round 1

Reviewer 1 Report

In my opinion, the article is interesting and well written.  Below there are some suggestions for Authors.

1. For greater clarity, I propose to change the title of section 2.3. on "Methodology"and justify a little more the desirability of particular types of tests.

2. In the introduction and in the conclusions, the Authors note that a novel strategy for designing uniform, stable and high-performance surfaces has also been developed. It would be good to present this idea in more detail in the section "Discussion"

Reviewer 2 Report

Authors should provide response to following queries and must incorporate the required changes at appropriate places in the Manuscript.

1.     Please specify the novelty of the work.

2.     What are the gaps in the literature that authors want to address?

3.     How the current approach is different and effective from already published work?

4.     In the result section, provide comparative study of the as observed results with respect to already reported.

Reviewer 3 Report

The manuscript, by Zhang et al., presents a detailed study on the surface-functionalization of Al-alloy powder by modification with an organic coating with the aim of its passivation and increase of the powder’s chemical resilience towards oxidation. This is done with the purpose of improving the possibilities for long term storage of this Al-based material, which finds application in energetic materials and explosives, and to prevent its reaction with oxygen and moisture which would otherwise have a detrimental effect on its energy content. The authors propose a two-step modification of an initial trimethoxysilane based layer (FAS-17) in order to passivate the surface, and then glycidyl azide polymer (GAP), which is also an energetic compound and improves the compatibility of the modified material with other components of the explosive mixture. The modification strategy is described in detail, and a number of characterisation techniques are used to establish its effects on the functionalized materials structure, morphology and physicochemical properties.

Overall, I am pleased with the quality of the manuscript - the experimental protocol seems logical, the use of characterisation techniques is motivated and linked well to the points that the authors want to prove. The quality of presentation is satisfactory and the text is readable and accessible (even though there seems to be an overuse of past tense). 

In general, my recommendation is that the paper is considered for publication in MDPI Materials, after some minor corrections. I have listed some questions and recommendations below. 

1) The title and keywords are too general and do not reflect well the contents of the manuscript. I realise that a change of the title is difficult at this point, but I strongly recommend that the keywords should be corrected - for example “chemistry/physics synergy” does not mean much.

2) The Introduction could be enhanced with a stronger focus on the reasons for- and examples of the surface funcionalization of Al powders and more references to literature works on similar passivation methods could be added.

3) Do the authors consider “W-Al” to be a proper notation for the material described in the manuscript ? I understand that it is “water-reactive-Al”, however, upon the initial read-through I was felt with an impression that it is a tungsten-aluminum alloy (especially given that the only mention about the Al-Ga-In-Sn composition is mentioned in section 2.1).

4) In the characterisation section it is mentioned that Au-sputtering was used for SEM preparation. Is this correct, since SEM-EDX is performed and one would expect carbon-coating ?

5) The notation in Figure 1 (Fig 1(A) / Fig 1(a)) is confusing. Probably the authors may consider shifting 1(A) as a separate figure in the Experimental section as a general scheme of the functionalization strategy and proposed mechanism.

6) Please add a bit of more details in the text, when referring to previous work. E.g.

* The preparation procedure of the W-Al powder is mentioned in Section 2.1. to be based on the “previous method [10].” Add a few lines describing the procedure (e.g., - “Briefly, the Al, Ga, In and Sn metals were mixed in a A:B:C:D molar/mass ratio and melted together at …. under Ar. The resulting casts were mechanically crushed … sieved… and … ball-milled to obtain powders of …as described in our previous work [10].” 

* Same with 2.3. Characterisation - “… water reactivity … was evaluated by measuring the volume of hydrogen production … based on our previous work [28]”. Obviously, the setup is quite dedicated, but just add a line or two to help the reader understand  

Round 2

Reviewer 2 Report

NA